# Advances in Targeted Therapy of Breast Cancer with Antibody-Drug Conjugate

**DOI:** 10.3390/pharmaceutics15041242

**Published:** 2023-04-14

**Authors:** Md Abdus Subhan, Vladimir P. Torchilin

**Affiliations:** 1Department of Chemistry, ShahJalal University of Science and Technology, Sylhet 3114, Bangladesh; 2Center for Pharmaceutical Biotechnology and Nanomedicine (CPBN), Department of Pharmaceutical Sciences, North Eastern University, Boston, MA 02115, USA; 3Department of Chemical Engineering, North Eastern University, Boston, MA 02115, USA

**Keywords:** ADCs, breast cancer, drug resistance, non-internalizing ADCs, targeted therapy

## Abstract

Antibody–drug conjugates (ADCs) are a potential and promising therapy for a wide variety of cancers, including breast cancer. ADC-based drugs represent a rapidly growing field of breast cancer therapy. Various ADC drug therapies have progressed over the past decade and have generated diverse opportunities for designing of state-of-the-art ADCs. Clinical progress with ADCs for the targeted therapy of breast cancer have shown promise. Off-target toxicities and drug resistance to ADC-based therapy have hampered effective therapy development due to the intracellular mechanism of action and limited antigen expression on breast tumors. However, innovative non-internalizing ADCs targeting the tumor microenvironment (TME) component and extracellular payload delivery mechanisms have led to reduced drug resistance and enhanced ADC effectiveness. Novel ADC drugs may deliver potent cytotoxic agents to breast tumor cells with reduced off-target effects, which may overcome difficulties related to delivery efficiency and enhance the therapeutic efficacy of cytotoxic cancer drugs for breast cancer therapy. This review discusses the development of ADC-based targeted breast cancer therapy and the clinical translation of ADC drugs for breast cancer treatment.

## 1. Introduction

ADC-based drugs span the gap between mAb and cytotoxic drugs used to enhance the therapeutic effectiveness of cancer therapy, including breast cancer. The four key components of an ADC are the targeting antigen, the monoclonal antibody (mAb), the payload, and the linker. ADCs consist of an anticancer mAb conjugated to a cytotoxic payload through an engineered chemical linker that enables the effective targeting of tumor cells and a killing effect simultaneously [1]. ADCs have combined characteristics of both chemotherapeutics and targeting agents by joining them with a linker. ADC drugs have shown anti-cancer effectiveness against human epidermal growth factor receptor 2-positive (HER2+) breast cancers as well as anti-cancer effects on HER2-low and HER2-negative breast cancer and in triple-negative breast cancer (TNBC) patients [2]. The HER2 is overexpressed in 20% of breast cancer patients. Before the availability of HER2-directed mAbs, the prognosis for HER2 breast cancer was significantly low. Several novel HER2-targeting mAbs have been fabricated to bind to the HER2 receptor with more specificity than trastuzumab or to have the ability to bind to additional epitopes to improve function or to prompt a greater immunologic response [3]. These ADCs are developed to selectively deliver the payload directly to the target site in tumors [4]. The treatment of HER2-positive breast cancer is being greatly enhanced by ADCs such as trastuzumab deruxtecan, ARX788, and ZW49 and are proficiently utilized to deliver the potent cytotoxic drugs by using mAbs with low off-target toxicities. Usually, ADCs are stable in the circulation, and they readily target and finally release their payloads in the vicinity of the cancer cells. Internalization and the intracellular transport mechanism of ADCs have a positive impact on the cytotoxicity of ADCs. Each component can affect the efficacy and safety of the ADC drug. Therefore, ADC development needs to consider all these components, including the choice of the target antigen, the mAb, the cytotoxic payload, the linker and the conjugation approach [1].

The vital step of ADC development is the selection of target antigens [5]. Once the ADC is administered, the antibody binds to the target antigen on the tumor cells and is internalized by the tumor cells. The optimal target antigen should be tumor-specific with a homogeneous expression pattern and high levels of expression, rapidly internalizing with minimal ectodomain shedding. The level of antigen expression critically affects the therapeutic index of ADC drugs as it defines the amount of the cytotoxic payload that will be internalized into the cancer cells. Therefore, selection of target antigen is crucial for the effective internalization of ADCs into target cells. In solid tumors, the correlation between surface antigen density and intracellular ADC concentration has an optimal linear relationship [5]. The payload is then released into the lysosome, where the linker is decoupled, depending on intercellular conditions such as low pH and proteosome-induced degradation. The cytotoxic payload may kill cancer cells as a result of DNA damage. Cancer cells may also be inhibited by microtubule disrupting agents. Some membrane permeable payloads (e.g., SN-38) can cross the cell membrane and exert a cytotoxic killing effect on bystander cancer cells, which may or may not express target antigens (Figure 1) [2,6]. The ADC internalization and extracellular pathway may have a crucial impact on the cytotoxicity of ADC [7].

Currently, more than 12 ADCs have received Food and Drug Administration (FDA) approval for hematological malignancies and solid tumors, and around 100 ADC drugs are currently in different phases of clinical trials [1,8]. Most of the ADCs are in different phases of clinical trials for solid tumors, including breast cancers (Figure 2) [1,9,10,11,12,13,14]. The majority (seven out of twelve) of the approved ADCs target non-solid tumors. All approved ADCs employ internalizing antibodies, and the majority of them utilize cleavable linkers. In 2000, the first ADC drug Mylotarg (gemtuzumab ozogamycin) was approved by the US-FDA for adults with acute myeloid leukemia (AML), which was the early approach for ADC-targeted cancer therapy. The ADC drug ado-trastuzumab emtansine or T-DM1 (Kadcyla) for HER2-positive metastatic breast cancer was FDA approved in 2013. Currently, the clinical advancement of ADC drugs for breast cancer therapy has emphasized three FDA-approved agents: trastuzumab emtansine, trastuzumab deruxtecan and sacituzumab govitecan.

In 2020, sacituzumab govitecan received accelerated approval for metastatic TNBC (triple-negative breast cancer) patients who have received at least two prior therapies for metastatic disease. In 2021, the FDA finally approved sacituzumab govitecan (Trodelvy by Immunomedics Inc., Morris Plains, NJ, USA) for patients with unresectable, locally progressed or metastatic TNBC who had experienced two or more prior systemic treatments, with at least one of them for metastatic disease. In 2022, another ADC drug, fam-trastuzumab deruxtecan-nxki (Enhertu), received FDA approval for unresectable or metastatic HER2-low breast cancer patients who had received prior chemotherapy for a metastatic disease or had developed disease relapse during or within six months of finishing adjuvant therapy. Around 40 to 50% of breast cancer patients have tumors with low HER2 expression. HER2-low breast cancer is defined as immunohistochemistry [IHC]1+ or IHC2+ with no HER2-amplification (in situ hybridization test negative, ISH–) [15]. In February 2023, the US-FDA approved sacituzumab govitecan-hziy for the treatment of patients with unresectable locally progressed or metastatic hormone receptor positive (HR+) and HER2-negative (IHC 0, IHC 1+ or IHC 2+/ISH–) breast cancer patients who have experienced endocrine-based therapy and at least two more systemic therapies for metastatic disease [6,16]. This approval of sacituzumab govitecan-hziy for (HR+) and HER2-negative breast cancer patients was based on the TROPiCS-02 (NCT03901339) trial.

Sacituzumab govitecan is composed of an antibody coupled to topoisomerase I inhibitor (SN-38) through a proprietary hydrolysable linker, and it targets human trophoblast cell-surface antigen 2 (TROP-2), which is overexpressed in many epithelial cancers, including a majority of breast cancers [6]. An overexpression of TROP-2 in breast cancer is allied with a poor prognosis and a low survival rate. TROP-2 is also known as tumor associated calcium signal transducer 2 (TACSTD2), which is a cell surface glycoprotein functioning as a transmembrane transducer of intracellular calcium signals. TROP-2 plays roles in cell proliferation, invasion, migration, apoptosis, and treatment resistance by binding to or interacting with several molecules. The TROP-2-targeting ADCs make TROP-2 an accessible and innovative therapeutic target for advanced metastatic cancer [17]. The FDA has accepted for review a biologics license application (BLA) for vic-trastuzumab duocarmazine (SYD-985) for the treatment of HER2-positive patients with unresectable, locally advanced or metastatic breast cancer [18].

ADC-based drugs signify a growing field of cancer therapy, including for breast cancer. Different ADC drug technologies advanced over the past decade have generated a variety of opportunities for engineered innovative ADCs [19]. Advances in the engineering of linkers and the appearance of novel payloads has led to progress for innovative ADCs for cancer therapy [1]. Prospective antigen targets were revealed for both solid and hematological tumors [20,21]. Many potential cytotoxic agents have been found, including microtubule inhibitors, anthracycline, auristatins, and maytansinoids [22,23,24,25,26] (Figure 3). Further, new generations of linkers have been introduced to improve the therapeutic applications of ADC drugs [19,21,27,28,29]. Bispecific antibodies have been designed to improve both the potency and specificity of ADCs to deliver multiple classes of payloads [30,31,32,33]. The majority of currently utilized ADCs use human immunoglobulin G (IgG) antibody, the most frequently utilized isotype of antibodies (found in serum) for cancer immunotherapy. Since the size of the mAb is larger compared to the cytotoxic payload (around 90% of mass of any given ADC) it may hamper the delivery of ADC-based drugs to target cells and, hence, may hamper their efficiency. Although this does result in a reduced distribution of cytotoxic payload to healthy tissues, including metabolizing and eliminating organs (liver, intestines, muscle, skin etc.), overall, it may affect the pharmacokinetics and pharmacodynamics as well the antitumor efficacy of ADCs. Small-size antibodies or fractions or alternatives may be utilized in ADCs to reduce these difficulties. There are many studies that utilize antibodies derived from IgGs or small binding proteins, small molecules, and peptides for ADC construction, such as antigen-binding fragments (Fab), single chain variable fragments (scFv), variable domains (VHH; known as nanobodies), diabodies, affibodies, knottin, DARpin, dAb, and bicyclic peptides (Figure 3). Because of their smaller size compared to regular IgGs, they demonstrate enhanced pharmacokinetics for tumor penetration [34,35,36]. Therefore, modified ADCs with smaller IgG alternatives or small-format drug conjugates may be an effective and innovative therapeutic for cancers, including breast cancer.

Furthermore, the combination approaches that utilize ADCs in clinical trials are being explored extensively, for example by combining them with immune checkpoint inhibitors (ICIs) and chemotherapeutics.

Thus, the development of new ADC drugs has offered huge opportunities for the treatment of cancer, including breast tumors [34]. The experience with existing ADCs has been useful in developing new generations of more effective ADCs, such as site-specific conjugations developed for the fabrication of ADCs to produce homogeneous ADCs with a drug-to-antibody ratio of 2 or 4 as well as those with improved pharmacokinetics [19].

Currently, ADC drugs require internalization to release the cytotoxic payload, which generates a hurdle for therapeutic development in that it requires cancer cell targets that overexpress internalizing antigens for effective intracellular processing [8].

However, an alternative strategy is emerging for the extracellular release of the payload from cleavable linkers: its binding to weakly internalizing antigens or other constituents of a tumor, such as secreted proteins in the tumor microenvironment (TME). This may eliminate the dependence on the overexpression of antigen, avoid inefficient internalization, and potentially improve the variety of cancer targets in the components of the extracellular tumor matrix for an enhanced therapeutic effect [8].

In addition to the well-recognized ADC targets, including HER2, EGFR, FGFR and c-MET, there are many emerging ADC target antigens, including TROP-2, CD-25, EpCAM, demonstrating promising outcomes in solid tumor therapy in preclinical and clinical settings (Table 1) [8,37]. For non-internalizing ADCs, secreted extracellular proteins and abundant stromal and vasculature components containing collagen, fibrin, fibronectin, and tenascin-C are emerging as prospective targets (Table 1) [8].

## 2. Internalizing and Non-Internalizing ADCs

The majority of the current ADC drugs utilize the internalizing mechanism. After the ADCs’ binding to the target antigen expressed on the cancer cell, the ADCs are internalized through endocytosis and subsequently degraded in the lysosome, and the cytotoxic payload is released (Figure 4A) [8]. The membrane permeable cytotoxic payload may pass out of the cell from which it was released and exert a bystander killing effect on surrounding tumor cells with or without antigen expression. In recent years, some internalizing ADCs have received FDA approval for clinical use. However, internalizing ADCs suffer from several drawbacks. The internalizing ADCs rely on a high expression of antigens, and many tumors lack such antigen expression. As antibodies are large proteins, they inadequately penetrate tumors due to sluggish diffusion. Furthermore, the antibodies’ continuous binding to cancer cells at the outer edge of the tumors close to blood vessels may prevent their dispersion and thus bind to cancer cells deeper within the tumor through the antigen barrier. Cancer cells also acquire resistance to antibodies due to alterations in several processes, including internalization, trafficking, antigen reuse, and a lysosomal degradation mechanism [38,39].

Non-internalizing ADCs may utilize an alternative mechanism of action. In non-internalizing ADCs, instead of necessitating lysosomal disintegration to release payload intracellularly, the ADC’s labile linkage may provide the release of the payload into the extracellular TME, targeting the distinct chemical/enzymatic atmosphere of the tumor, and they may disperse into neighboring cancer cells to exert their cytotoxic effect (Figure 4B). Non-internalizing ADCs do not need antigen overexpression on cancer cells, and may target several physiological features common to most of the aggressive cancers. The extracellular release of the payload may also facilitate a deeper diffusion of the cytotoxic drugs into tumors due to easy penetration, and it may exert a bystander effect. Non-internalizing ADCs may be suitable for targeting cancers with dense tumor stroma, such as TNBC and pancreatic ductal adenocarcinoma (PDAC). Therefore, a non-internalizing ADC may serve as a broad-spectrum anti-cancer therapy.

Internalizing ADCs usually utilize non-cleavable linkers, which involve the mAb degradation of their integral amino acids for the release of a cytotoxic payload into the lysosome. The charged nature of attached amino acids render it incapable of dispersing across the cell membrane and inhibit nearby cancer cells through a bystander effect. ADCs that use non-cleavable linkers necessitate effective cellular trafficking. Non-cleavable linkers are effective for internalizing ADCs. Among the currently approved ADCs for breast cancer therapy, only trastuzumab emtansine possesses a non-cleavable linker (Figure 3). Cleavable linkers have a chemical trigger that are dissociated in the specific location of tumors to release the cytotoxic payload. Many cleavable linkers are currently utilized in ADCs, such as acid-labile hydrazones, reducible disulfides and enzyme-cleavable linkers (e.g., valine-citrulline). Several FDA-approved breast cancer ADCs, including sacituzumab govitecan, and trastuzumab deruxtecan, have cleavable linkers (Figure 3). Most of the cleavable linkers are susceptible to intracellular cleavage. However, chemical and enzymatic cleavage triggers may exist extracellularly [18,19]. Further, expiring tumor cells may release high concentrations of intracellular species (including glutathione and protease) into the TME, which facilitate the development of non-internalizing ADCs. Non-internalizing ADC mechanisms may involve targeting membrane proteins, the TME and the tumor stroma or vasculature. As a result, non-internalizing ADCs provide a wider choice of antigen targets, avoid insufficient internalization and trafficking to the lysosome, improve the bystander effect, and enhance tumor penetration to deeper cancer cells (Figure 4B). Internalizing ADCs’ tumor cells can acquire resistance mechanisms. However, non-internalizing ADCs may reduce the drug resistance.

Internalization is key to the effectiveness of ADCs with cleavable linkers. However, increasingly, several indications suggest a contrary mechanism. The dying tumors may release the intracellular linker triggers into the extracellular TME, facilitating the non-internalizing mechanisms of ADC. These linkers may be incorporated into non-internalizing ADCs. Therefore, the concept that the linker must persist in circulation before internalization and detachment within the cancer cells may need to be modified for non-internalizing ADCs. The linkers comprising ester or carbonate groups with modest stability in circulation have been utilized in integration with antibodies targeting the stromal constituents in the TME, permitting a persistent local drug release into tumors. Carbonate linkers already have been proven in a clinical setup (Sacituzumab govitecan). This linker type may be utilized only with moderately cytotoxic payloads. However, off-target toxicity due to labile linkages should be investigated [8].

ADCs’ effectiveness depends on several features of target antigens [2,5]. For internalizing ADCs, the overexpression of target antigens on the cancer cells is necessary. However, no or low-level expression on healthy cells is needed. The landscape of antibody–antigen interactions may dictate the extent of the internalization of the ADCs. For the effective internalization of ADCs, the antibody must prompt rapid receptor internalization, intracellular trafficking, and lysosomal degradation. On the other hand, non-internalizing ADCs may not require a substantial rate of internalization after antibody binding because these intracellular steps are not significant for the non-internalizing ADCs’ mechanism [8]. The pros and cons of internalizing and non-internalizing ADCs is summarized in Table 2.

## 3. Development of ADC Drugs for Targeted Breast Cancer Therapy

Breast cancer is the most frequently diagnosed cancer in women, which is the second most prevalent cause of death among cancers. In 2020, 11.7% of newly diagnosed cancer cases were for breast cancer, which significantly endangered women’s health worldwide and was the main cause of female death [7,40].

Currently utilized therapies for breast cancer are chemotherapy, radiation therapy and surgical resection. These options are associated with severe side effects, therapy resistance and disease recurrence. In metastatic disease, therapy options are more limited, and a majority of patients will die from the disease. In recent years, targeted therapy using ADCs have greatly advanced the treatment of breast cancer by utilizing HER2-targeted, HR-targeted, and TNBC-targeted ADCs [41].

### 3.1. Human Epidermal Growth Factor Receptor-Targeted ADC Therapy for Breast Cancer

#### 3.1.1. HER2-Targeted ADCs

HER2-targeted therapies of breast cancer have shown promising results in clinic. Some of the HER-2 targeted therapies have been approved by FDA (Trastuzumab emtansine, Trastuzumab deruxtecan), and some others are in late stages of clinical trials (Trastuzumab Duocarmazine, Disitamab Vedotin etc.).

#### 3.1.2. Trastuzumab Emtansine

Trastuzumab emtansine (TDM-1) was approved for the treatment of patients with unresectable or metastatic HER2-positive breast cancer. The approval was based on the EMILIA trial for patients who received trastuzumab and taxane or reverted during the therapy or within 6 months following adjuvant therapy [42]. DM1 is a highly potent cytotoxic agent for inhibiting tubulin polymerization and causes death in proliferating cancer cells [1,6,43]. The TDM-1 ADC drug was approved as a monotherapy and is utilized as second-line therapy.

The KATHERINE trial supported TDM1-1 as the standard adjuvant treatment for patients with HER2-positive breast cancer [44]. The development of TDM-1 as a first-line therapy for HER2-positve metastatic breast cancer or as a neoadjuvant therapy has not yet been shown.

T-DM1 remains a second-line standard therapy for HER2-positive breast cancer patients. According to the phase III MARIANNE trial, the TDM-1 with or without pertuzumab demonstrated no improvement in PFS as a first-line treatment for metastatic HER2-positive breast cancer [42]. The CLEOPATRA trial showed that pertuzumab added to taxane and trsatuzumab improved both PFS and OS [45,46,47]. The neoadjuvant KRISTINE study demonstrated that docetaxel, carboplatin and trastuzumab plus pertuzumab improved pCR in a significant number of patients compared to the T-DM1 plus pertuzumab arm. The CompassHER2-RD study (NCT04457596) assessed whether the integration of T-DM1 with tucatinib was better than TDM-1 alone in metastatic breast cancer patients pretreated with trastuzumab plus taxane.

#### 3.1.3. Trastuzumab Deruxtecan

Trastuzumab deruxtecan was approved based on the results of the DESTINY-Breast01 study. This approval is for breast cancer patients with HER2-positive unresectable or metastatic disease who received a minimum of two previous anti-HER2 therapies [1]. T-Dxd is made of a humanized anti-HER2 mAb and a cleavable tetrapeptide-based linker attached to a potent cytotoxic topoisomerase I inhibitor, an exatecan derivative. T-Dxd possesses a high drug-to-antibody ratio (DAR) (8), which facilitates the release of a high concentration of payload [48]. However, DAR is one of the crucial factors in determining the efficacy of ADCs. DAR values are important for the therapeutic index of ADCs. A low DAR (i.e., few drug molecules on each antibody) may cause reduced efficacy, while a high DAR may affect ADC structure, stability, and antigen binding, resulting in poor pharmacokinetics for ADC drugs because of higher hydrophobicity and lower solubility [34]. The DAR of the majority of the ADCs currently in clinical trial are in the range of 2 to 4. Hence, controlling the DAR of ADCs during ADC preparation is a key process, and the real-time DAR analysis of in situ ADC drugs may be an effective approach. Currently, there are several methods for DAR measurement, including UV-visible spectroscopy, hydrophobic interaction chromatography, RP-HPLC, and LC-MS. The antibody deglycosylation of ADC samples simplifies DAR measurement. Although deglycosylation is time consuming, an optimal deglycosylation of ADCs and a rapid LC-MS analysis for DAR detection with real-time monitoring may be an effective approach to control and optimize the DAR of ADCs in clinic [49].

Upon binding to HER2, T-Dxd is internalized and trafficked intracellularly to the lysosome [50]. The linker is stable in plasma and experiences a specific breakup by lysosomal cathepsins, which are overexpressed in cancer cells [51,52,53,54]. The payload can readily cross the cell membrane (~5 nm) due to its membrane-permeable nature, and it exerts an effective cytotoxic effect on bystander cancer cells, irrespective of HER2 expression levels [55].

Both the in vitro and in vivo pharmacological function of T-Dxd was evaluated and compared to T-DM1 using patient-derived xenograft (PDX) models [54]. The highest non-severe toxic dose of T-Dxd was 30 mg/kg in cynomolgus monkeys [54]. The study also evaluated the efficiency of T-Dxd in T-DM1-unresponsive PDX-models with HER2 expression (high and low) [55]. T-Dxd demonstrated a bystander cancer-cell-killing effect in addition to direct cytotoxicity. The T-Dxd cytotoxic payload is strongly membrane-permeable compared to T-DM1. T-Dxd demonstrated low or no systemic toxicity due to a bystander effect.

T-Dxd was investigated in several clinical trials for solid tumors focusing on breast cancer. In 2015, a phase I first-in-human trial included highly treated patients with progressed HER2-positive breast cancer [48]. Breast and other cancer patients were enrolled at two locations in Japan. The highest tolerable dose was not achieved based on primary tumor inhibition and safety data obtained in this trial. The recommended dose was selected as 5.4 to 6.4 mg/kg body weight. In another phase I study, patients with HER2-positive progressed breast cancer who had previously received T-DM1 therapy were examined in a T-Dxd trial at fourteen centers, including eight in the USA and six in Japan [51]. A phase Ib study assessed the effectiveness of T-Dxd in HER2-low progressed or metastatic breast cancer when treated with the recommended dosage [52]. Heavily pretreated patients were enrolled for this study, and 37% (20/54) demonstrated a confirmed radiological response. Treatment-related adverse effects of grade 3 or more were detected in 63% (34/54) of patients including a low neutrophil count, low WBC count, anemia, hypokalemia, low platelet count, aspartate aminotransferase (AST) high, low apatite, febrile neutropenia, cellulitis and diarrhea with three fetal effects due to interstitial lung disease (ILD). The preliminary clinical studies of T-Dxd demonstrated prospective anticancer activity in both HER2-positive and HER2-negative breast cancer patients, with a significant percentage of patients suffering modest to serious side-effects, particularly with T-Dxd allied potential ILD.

T-Dxd received FDA approval in 2020 [56]. In 2019, the primary outcomes of the DESTINY-Breast01 trial (NCT0348492) were issued [6,56]. This phase II, two-part, single group, multicenter trial of T-Dxd assessed T-DM1 patients with HER2-positive metastatic breast cancer. After a median follow up of 11.1 months, at the recommended dose (5.4 mg/kg) of T-Dxd, 60.9% (112/184) of patients showed a confirmed radiological response. The median number of previous lines of therapy for metastatic patients was 6 months. The median PFS was 16.4 months. The commonly observed side effects of grade 3 or more were neutropenia in 20.7% (38/184) patients, anemia in 8.7% (16/184), and nausea in 7.6% (14/184). T-Dxd-allied ILD was observed in 13.6% (25/184) of patients experiencing any-grade ILD. In December 2019, an update of the study demonstrated a generally tolerable safety profile with the formerly finished outcomes [57]. In the follow up period, five patients (2.7%) died due to the ILD. T-Dxd received accelerated FDA approval in December 2019 due to this study outcome. T-Dxd also received approval from the European Medicine Agency (EMA) and Swissmedic, in the meantime.

The DESTINY-Breast02 study is a phase III, open level, randomizing one of two arms, T-Dxd versus treatment of physician choice (TPC) (a combination of capecitabine with either trastuzumab or lapatinib) trial for HER2-positive metastatic breast cancer patients previously treated with T-DM1. Fam-trastuzumab deruxtecan-nxki led to higher response rates and longer survival times in a third-line setting for HER2-positive metastatic patients pretreated with T-DM1. According to the result form the DESTINY-Breast02 phase III study presented at the 2022 San Antonio breast cancer symposium (Abstract GS2-01), among the 608 patients enrolled and treated with T-Dxd, 69.7% experienced an objective response compared to TPC (29.2%). Those patients treated with T-Dxd were 64% less likely to experience disease progression than the patients that received TPC. The PFS was 17.8 months for T-Dxd-treated patients versus 6.9 months (TPC). The OS was 39.2 months for the patients treated with T-Dxd versus 26.5 months with TPC [58].

The outcomes of the DESTINY-Breast03 trial were presented in 2021 [6,41]. In this phase III trial, T-Dxd was evaluated and compared to T-DM1 in breast cancer patients pretreated with trastuzumab and taxane. The 12-month PFS was 75.8% for T-Dxd and 34.1% for T-DM1. The OS was 94.1% for T-Dxd and 85.9% for T-DM1. The median therapy duration was favorable with T-Dxd compared to T-DM1. The second-line treatment with T-Dxd led to a noticeably longer OS compared to T-DM1 in patients with HER2-positive metastatic breast cancer, according to updated results from the DESTINY-Breast03 phase III study presented at the San Antonio Breast Cancer Symposium on 6–10 December 2022. The DESTINY-Breast03 study compared the efficacy and safety of T-Dxd with those treated with T-DM1 in patients with HER2-positive metastatic breast cancer that advanced on or after first-line treatment. This study confirmed the benefit of T-Dxd for PFS and the substantial improvement in OS. Future analyses of the DESTINY-Breast03 trial may investigate the efficacy of T-Dxd in patients with brain metastasis to discover predictive markers of response. Further, ongoing studies aim to determine the efficacy and safety of T-Dxd as first-line therapy for patients with HER2-positive metastatic breast cancer. A shortcoming of the trial was the disproportionate enrollment of Asian patients compared to the US and European patients. An additional restriction was that the median OS was not reached at the time of this analysis [59].

T-Dxd has been allied with higher rates of toxicities, including grade 5 events [60]. The study demonstrated that Japanese patients were more susceptible to growing ILD after therapy with T-Dxd than those of other origins. According to the DESTINY-Breastcancer01 trial, the median time to onset of treatment-related lung diseases was 6 months. Monitoring the patient’s signs and symptoms related to lung disease and consulting with pulmonologist may be helpful during T-Dxd therapy [61]. Further studies are required to assess the best tracking and handling of ILD associated with T-Dxd treatment.

#### 3.1.4. Trastuzumab Duocarmazine

Trastuzumab duocarmazine (SYD985) is an anti-HER2 ADC composed of anti-HER2 mAb with the same sequence of trastuzumab, conjugated to a cleavable linker comprising duocarmycin. Duocarmycin results in DNA degradation in cancer cells, eventually resulting in cancer cell death [62]. Proteases, such as cathepsin B secreted by tumor cells, could be active extracellularly, which would facilitate a bystander killing effect [63]. Trastuzumab duocarmazine demonstrated prospective preclinical cancer cell killing efficiency in breast tumors with different HER2 expression levels (either low or high) [64].

The results of a first-in-human, phase I clinical trial of trastuzumab duocarmazine evaluated dose-escalation and expansion (NCT02277717) in 146 patients with HER2-positive locally progressed or metastatic solid tumors, refractory to standard cancer therapy [65]. A prospective ORR of 33% (16/48) was observed (n = 48) according to response evaluation criteria in solid tumors (RECIST). Among the 47 evaluable patients, ORR was 28% (9/32) among the patients with HER2-positve and HER2-low and 40% (6/15) among the patients with HER2-negative and HER2-low, respectively.

The phase III TULIP trials included the evaluation of the PFS of trastuzumab duocarmazine versus TPC (NCT03262935). The preliminary result of the TULIP study was reported at the 2021 ESMO meeting [66]. In this study, which enrolled 437 patients, the median PFS was 7 months. Trastuzumab duocarmazine may be a promising therapy for HER2-positive breast cancer patients with heavily pretreated, locally progressed or metastatic disease.

#### 3.1.5. Disitamab Vedotin

The ADC drug disitamab vedotin (RC48-ADC) consists of an mAb (disitamab) that targets HER2 and a cleavable linker conjugated to the microtubule-disrupting, synthetic antineoplastic agent monomethyl auristatin E (MMAE). Investigations have demonstrated that RC48-ADC inhibits tumor cells, effectively targeting HER2 [67]. Superior tumor inhibition of disitamab vedotin compared to T-DM1 was exhibited by xenograft models [67]. In a phase I trial, disitamab vedotin demonstrated good tolerability and promising efficiency (ORR 46.7%) in locally progressed or metastatic HER2-positive breast cancer patients [68]. A randomize phase II trial is currently assessing the efficiency of disitamab vedotin 2 mg/kg given every 2 weeks (NCT03500380). A randomized phase III trial is currently assessing the efficacy and safety of disitamab vedotin versus TPC in patients with low HER2-expressing metastatic breast cancer who experienced disease progress during or after one line of therapy for metastatic disease (NCT04400695) [1,6,68].

#### 3.1.6. Sachituzumab Govitecan in HR+ and HER2-Negative Breast Cancer Therapy

The US-FDA has approved Sacituzumab govitecan-hziy for the treatment of patients with unresectable locally progressed or metastatic hormone receptor positive (HR+) and HER2-negative breast cancer. This approval is for breast cancer patients who experienced endocrine treatment and a minimum of two more systemic therapies for metastatic disease. Sacituzumab govitecan is comprised of an antibody coupled to a topoisomerase I inhibitor through a hydrolysable linker. This is the first approved ADC drug targeting TROP-2 for the treatment of metastatic HR+ and HER2-negative breast tumors [1,6,18]. This approval for Sacituzumab govitecan-hziy is based on the phase III TROPiCS-02 trial. Sacituzumab govitecan-hziy is now recommended by the national comprehensive cancer network (NCCN) guideline. The TROPiCS-02 is a global, multicenter study comparing Sacituzumab govitecan-hziy to TPC [1,6,69].

The TROPiCS-02 trial demonstrated an OS benefit of 3.2 months from Sacituzumab govitecan-hziy compared to TPC (single agent chemotherapy, median OS:14.4 months versus 11.2 months) for patients with metastatic HR+ and HER2-negative breast tumors [1,6]. In this study, Sacituzumab govitecan-hziy established a 34% reduction in disease progression risk or death. The median PFS was 5.5 months versus 4 months. Recently, sacituzumab govitecan-hziy received FDA approval for patients with metastatic HR+ and HER2-negative breast cancer [16].

#### 3.1.7. Other ADCs Targeting Human Epidermal Growth Factor Receptor

HER3 is upregulated in metastatic breast cancer and other cancers, and it has been associated with poor outcomes [70,71]. The ADC drug patritumab deruxtecan (U3-1402) consists of human anti-HER3 IgG1 mAb conjugated to a topoisomerase I inhibitor via a cleavable linker. The assessment of the NCT02980341/JapicCTI-163401trial exhibited prospective tumor inhibition activity in heavily pretreated patients with HER3-expressing metastatic breast cancer [72,73,74].

ARX-788 is an innovative ADC consisting of anti-HER2 mAb, a noncleanable linker and a proprietary of MMAF (Amberstatin 269 or AS269). ARX-788 demonstrated more efficacy than T-DM1 for the inhibition of breast tumors resistant to T-DM1 [2,75]. ARX-788 is effective against breast cancer and other cancers. An ARX-788 ADC drug is currently undergoing two phase I clinical studies (NCT02512237, NCT03255070). Further, a different phase II clinical study is also assessing ARX-788 for HER2-positive metastatic breast tumor therapy (NCT05018676), HER2-mautated or HER2-overexpressed cancers (NCT05041972), HER2-low breast tumors (NCT05018676), and HER2-positive breast cancers with brain metastasis (NCT05018702) [75].

The A166 ADC drug consists of an anti-HER2 antibody and an effective MMAF-based payload (duostatin-5) conjugated via a cleavable valine-citrulline linker [76]. A phase I/II clinical trial of A166 ADC demonstrated clinical efficacy in patients with recurrence or progressed cancers. Patient’s responses were detected at the dose level of 3.6 mg/km and 4.8 mg/kg. An ORR of 36% was found at the effective dose (NCT03602079).

The ADC drug MRG002 consists of an anti-HER2 IgG1 mAb, a microtubule- disrupting agent, and MMAE, and is conjugated through a valine–citrulline cleavable linker [77]. This ADC drug exhibited a prospective breast tumor inhibition in patient-derived xenograft mouse models with a varying level of HER2 expression in preclinical study. Better MRG002 also exhibited higher cytotoxic potential compared to trastuzumab or T-DM1 in a xenograft mouse model. Further, combinations of MRG002 with an anti-PD-1 antibody showed noteworthy tumor inhibition activity. A phase I trial of MRG002 monotherapy is ongoing for breast cancer (CTR20181778, NCT04941339). Currently, several phase II studies with MRG002 are ongoing, evaluating its efficacy in multiple HER2-positive or HER2-low cancers [78].

Another innovative ADC drug, ALT-P7 (HM2-MMAE), is composed of trastuzumab biobetter HM2 conjugated to the payload MMAE [79]. A phase I study is evaluating the ALT-P7 ADC drug with HER2-positive breast tumor patients (NCT03281824). ALT-P7 exhibited a safety profile tolerable with dose-limiting toxicities found at 4.8 mg/kg and 4.5 mg/kg [80].

The HER-directed ADC drug ADC GQ1001 is a novel ADC drug with a trastuzumab mAb and DM1 cytotoxic payload, currently undergoing phase I/II clinical trial (NCT04450732) for HER2-positive tumors [2]. The trttuzumab zuvotolimod (SBT6050) ADC drug with an anti-HER2 mAb and toll-receptor 8 agonist payload is currently undergoing a phase I/II clinical trial (NCT 04460456) to asses anti-cancer activity against solid tumors, including breast cancer, as monotherapy and in combination with PD-1 inhibitors (such as pembrolizumab and cemiplimab) [2,81]. The SBT6050 developer is Silverback Therapeutics. SBT6050 targets the pertuzumab binding domain of HER2 and is designed to be utilized in combination with standard-of-care agents, including trastuzumab-containing regimens. Further, XMT-1522 ADC is currently undergoing phase I/II clinical trial (NCT02952729) for HER2-expressing progressed breast cancer therapy. This ADC drug is composed of mAb HT19, a polymer linker and a dolaflexin payload with DAR 12 [6]. ZW49 ADC with a N-acyl sulfonamide auristatin payload and cleavable linker is in phase I clinical trial (NCT03821233) for HER2-positive metastatic breast tumor. Another ADC drug targeting HER2 with a cytotoxic payload, a Tubulysin-based microtubule inhibitor, a cleavable linker, and a bystander cell-killing effect is in phase I/II clinical trial (NCT02576548).

### 3.2. Triple-Negative Breast Cancer-Targeted ADC Therapy

#### 3.2.1. Sacituzumab Govitecan in TNBC Therapy

TNBC is allied with a poor prognosis and has limited therapy options due to the missing of targets such ER, PR and HER2. TROP-2 is a calcium signal transducer found in a majority of epithelial carcinomas. It is overexpressed in breast cancer and TNBC tumors. TROP-2 is expressed in healthy tissues as well, with low expression. The ADC drug Sacituzumab govitecan targeting TROP-2 in TNBC cells has received approval from the FDA for TNBC therapy [82]. Sacituzumab govitecan consists of anti-TROP-2 mAb hRS7 IgG1κ and a cleavable linker CL2A conjugated to the SN-38. SN-38, an active metabolite of irinotecan, inhibits topoisomerase I and causes cell death due to DNA breaking by hindering the repair of DNA strands [77,83]. Sacituzumab govitecan demonstrated substantially increased tumor reduction compared to irinotecan in a TNBC-xenograft mouse model [84].

Sacituzumab govitecan received accelerated FDA approval for TNBC patients in 2020. In 2021, Sacituzumab govitecan received regular approval by FDA for patients with unresectable locally progressed or metastatic TNBC. This approval is for patients who experienced two or more prior systemic therapies, with a minimum of one of them for metastatic disease. Sacituzumab govitecan was first investigated by Bardia et al. in the IMMU-132-01 trial (NCT01631552), which was a phase I/II basket study that included 108 patients with advanced epithelial cancer who experienced a minimum of two previous therapies for metastatic TNBC [85]. The outcome of the study with metastatic TNBC patients was published in 2019 [86]. According to the study outcome, the ORR was 33% (34.3% BIR). The median OS was 13 months, and the median PFS was 5.5 months. The most common grade 3/4 adverse effects were observed in 42% of the patients (45/108) and included febrile neutropenia in 8% (9/108). Thirty-six TNBC patients experienced a radiological response, including thirty-three patients with a partial and three patients with a complete response. Finally, Sacituzumab govitecan demonstrated durable responses in patients heavily pretreated for metastatic disease.

The result of the phase III ASCENT trial led to the final approval of Sacituzumab govitecan by the FDA [87]. In a biomarker analysis of the ASCENT study, the efficacy of SG was evaluated [88]. The study showed that the clinical benefit of Sacituzumab govitecan versus TPC was irrespective of the level of TROP-2 expression or *BRCA1/2* mutation status. Patients with a high or medium TROP-2 expression level got an advantage from Sacituzumab govitecan therapy versus TPC in terms of ORR, PFS and OS. Because of the low number of patients with low TROP-2 expression, a definitive conclusion was difficult to make. Another subgroup analysis of TNBC patients with brain metastasis in the ASCENT study (n = 61) demonstrated better efficacy for Sacituzumab govitecan compared to TPC [89]. Therefore, the ASCENT study showed Sacituzumab govitecan as a later-line systemic therapy.

#### 3.2.2. TNBC Therapy with Ladrituzumab Vedotin

Ladrituzumab vedotin (SGN-LIV1A) is a novel humanized IgG1 ADC targeting LIV-1. LIV-1 is expressed frequently in breast cancer and is also expressed in a variety of cancer types. Ladrituzumab vedotin mediates the delivery of monomethyl auristatin E (MMAE), which facilitate anticancer activity through cytotoxic tumor cell killing and prompts immunogenic cell death (Figure 5) [90]. Ladrituzumab vedotin targets LIV-1 and is connected to MMAE through a cleavable linker. LIV-1 expression is observed in both ER-positive breast cancer and TNBC [91,92].

Preclinical studies demonstrated that ladrituzumab vedotin binds explicitly to the extracellular domain of LIV-1. After internalization, ladrituzumab vedotin releases its payload by proteolysis in the lysosome and results in the inhibition of microtubulin and prompts apoptosis [91].

A phase I study evaluated the safety, acceptability, pharmacokinetics, and tumor inhibition activity of ladrituzumab vedotin in patients with LIV-1-positive, unresectable, locally progressed or metastatic breast cancer who had a minimum of two or more cytotoxic treatments; ladrituzumab vedotin was administered every 3 weeks (NCT01969643) [93]. The outcome of the study was published in 2021 [94].

In an I-SPY study (NCT0102379) ladrituzumab vedotin was evaluated in patients with stage II/III breast cancer with neoadjuvant therapy with pCR as a major endpoint compared paclitaxel weekly for 12 weeks [95]. This study demonstrated no preeminence of the arm with ladrituzumab vedotin over the control with regard to assessed pCR [95]. Both ADCs and ICIs are emergent in the development of TNBC therapy. Therefore, the integration of ladrituzumab vedotin with pembrolizumab (NCT03310957) and with atezolizumab (NCT03424005) was evaluated in a phase Ib/II study. The preliminary data for the combination of ladrituzumab vedotin and pembrolizumab demonstrated that of 51 patients evaluated for safety and tolerability, 44 patients (86%) who received a PRP2D (preliminary recommended phase 2 dose) of 2.5 mg/kg reported treatment-allied adverse effects, including nausea [63]. The most frequent grade 3 or higher adverse effects observed in (5%) patients including neutropenia. In the 26 patients that were evaluated for efficacy and followed for at least 3 months, the ORR was 54%. Finally, the trial demonstrated that the combination regimen had an acceptable toxicity profile and prospective efficiency in metastatic TNBC patients.

## 4. ADCs Emerging as Promising Therapeutics for Breast Cancer Patients in Clinical Settings

As of February 2023, a total of twelve ADCs have received approval for different cancers, including three for breast cancer and TNBC therapy. Currently, more than 100 ADCs are in different phases of clinical trial. Among them, at least 13 are being evaluated for breast cancer therapy. ADCs targeting either HER or TNBC are potential therapeutics for breast cancer patients. The development of ADCs in the last decade has revolutionized breast cancer therapy [96,97,98,99,100,101,102]. Currently, two HER-directed ADCs have received FDA approval for HER2-positive metastatic breast cancer patients: ado-trastuzumab emtansine, trastuzumab deruxtecan. Very recently, sacituzuma govitecan has received FDA approval for HER2-negative metastatic breast cancer. On the other hand, sacituzumab goviteacn was previously approved by the FDA for TNBC patients [1,6,8]. There are also several ADC drugs in different phases of clinical trial for breast cancer patients (Table 3). Clinical and translational strategies may play a crucial role in enlightening the therapeutic windows for ADCs [103,104,105]. Although some potential ADCs used as monotherapy have demonstrated prospects in the clinical setting, combination approaches may be more effective to improve drug efficacy and reduce drug resistance in the clinical setting [102,106]. Further, clinical biomarkers to improve the choice of patients and the monitoring of response indication are also crucial for the enhancement of the therapeutic index of ADC drugs.

## 5. Conclusions and Future Perspective

ADCs, either targeting human epidermal growth factor receptor or TROP-2, are prospective treatment approaches for breast cancer patients. The growth of ADC-based drugs in the last decade has greatly enhanced breast cancer therapy. Three HER2-targeted ADCs have received approval for HER2-positive metastatic breast cancer patients, and one ADC drug was approved by the FDA for TNBC therapy. There are also many promising ADC drugs for breast cancer patients undergoing different phases of clinical trials. Ways to enhance the cancer cell uptake of ADCs have been key and have been allied with ADC development because most of the current ADCs depend on the expression of target antigens on tumor cells. An optimal patient selection approach necessitates the consideration of the intratumoral heterogeneity of the target antigen and dynamic changes in antigens associated with treatment and disease stages and optimizing the threshold of target antigen expression to improve ADC therapy in the clinical setting. Systemic toxicity is one of the major factors for the failure of ADCs in the clinical setup. Following ADC administration, the released payload rapidly appears within systemic circulation, which may cause systemic toxicity. Plasma exposure to released payload relates, partly, to the early deconjugation of the payload in circulation due to insufficient linker stability. Cleavable ADC linkers are frequently hydrolyzed at significant rates, leading to the premature release of the cytotoxic payload in the extra-tumoral compartments. Lipophilic payloads demonstrate high permeability through plasma membranes. Consequently, the free payload enters non-targeted cells via membrane diffusion, resulting in unwanted cytotoxicity. Approximately 0.1% of the administered ADC drugs are delivered to targeted tumor cell populations, with the vast majority of the injected dose catabolized ‘off-site’ within non-targeted healthy cells, leading to a potential unwanted toxicity. Off-site ADC toxicity may be of two types: on-target and off-target. On-target toxicity is due to the ADC binding to targeted cell surface protein of healthy cells, and off-target toxicity is due to non-targeted ADC binding [107].

The increasing attrition rate of ADCs with antimitotic payloads is due to suboptimal efficacy at the maximum tolerated doses (MTDs) during clinical trial. Recent development of ADCs demonstrated the utilization of more potent DNA-damaging payloads. Highly potent ADCs often lead to high efficacy; however, their efficacy is hindered by their life-threatening toxicities. Therefore, the unwanted toxicity limits the clinical benefit of ADCs by restricting the tolerable doses to amounts below the levels required to provide optimal anticancer effect. Studies have demonstrated that for many ADCs, the exposures at the clinical MTDs were much lower than the exposures essential for efficacy in preclinical models. The approaches to extend the therapeutic windows of ADCs through the attenuation of ADC toxicities might permit enhanced MTDs and, afterward, better clinical results [107].

Hepatotoxicity is included in one of the black box warnings for T-DM1 treatment, presenting as asymptomatic transient increases in serum transaminase. Black box warnings were added to the Sacituzumab govitecan label for severe or life-threatening neutropenia and severe diarrhea. Further, ILD and pneumonitis are included in the black box warning for patients treated with T-Dxd [107]. Toxic effects may be connected to several factors, including antibody, payload, linker and target antigen expression levels, and payload internalization mechanisms. Utilizing the bystander tumor cell killing effect, a proper pairing of cytotoxic payload, modification of properties of antibodies and linkers through the use of innovative conjugation technologies and drug/linker chemistry, regulation and optimization of DAR, utilization of bi-specific antibodies, non-internalizing ADC-based alternate mechanisms of action, and combination therapy strategies should be explored to overcome ADC drug resistance and enhance the effectiveness and the safety profile of ADC drugs. Clinical and translational policies may have an important role in increasing the therapeutic spectrum of ADCs. Clinical biomarkers to augment choice of patients and monitor the response signal are also crucial for advancing the therapeutic index of ADC drugs for effective breast cancer therapy and reduced mortality.

## Figures and Tables

**Figure 1 pharmaceutics-15-01242-f001:**
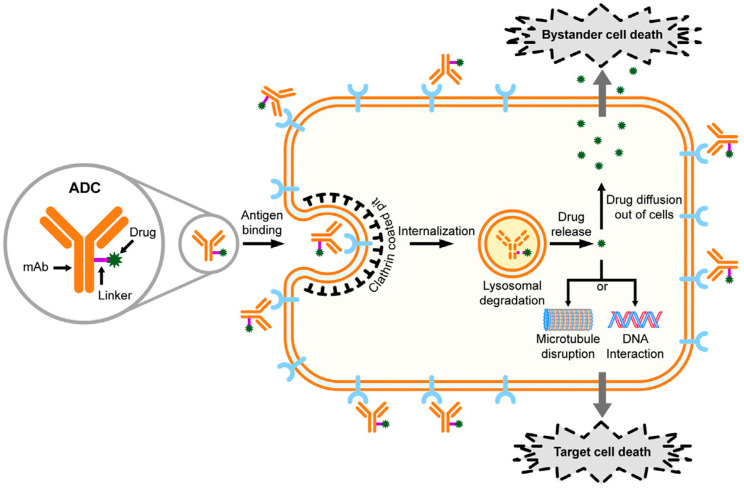
Mechanism of action of ADC drugs. Adapted from [2].

**Figure 2 pharmaceutics-15-01242-f002:**
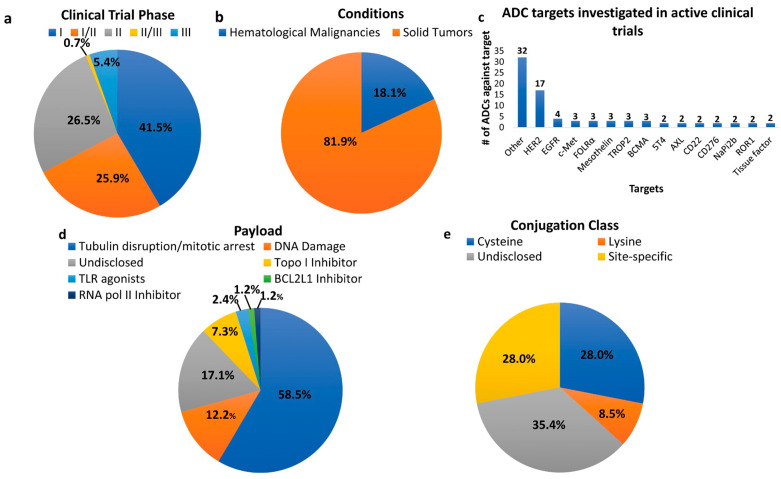
ADCs in clinical trials for cancer therapy. Adapted from [9].

**Figure 3 pharmaceutics-15-01242-f003:**
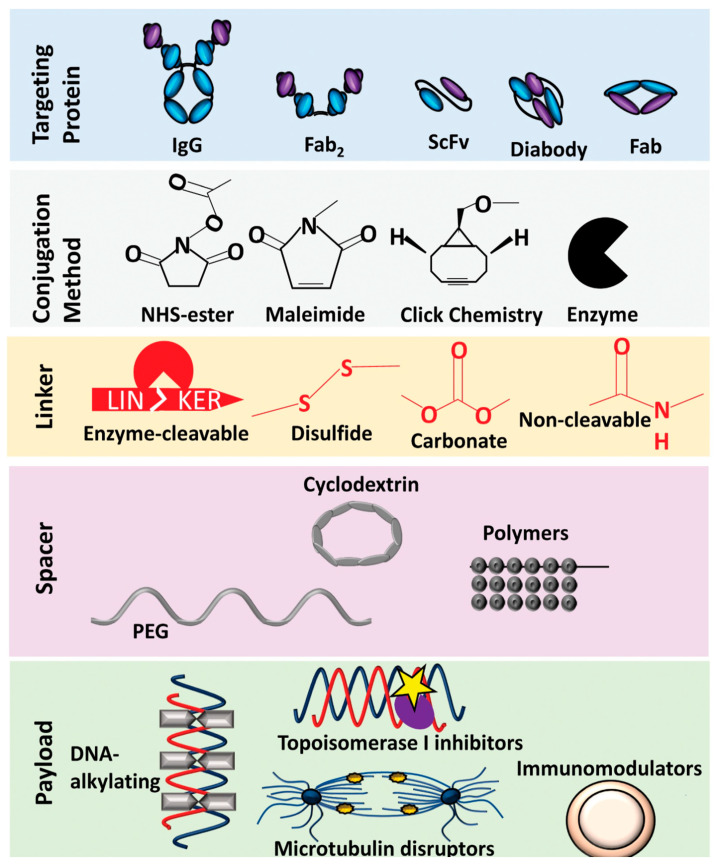
Growing the ADC framework through the utilization of new mAb, conjugation methods, linkers and spacer systems to optimize safety and effectiveness of ADCs for cancer therapy. Adapted from [9].

**Figure 4 pharmaceutics-15-01242-f004:**
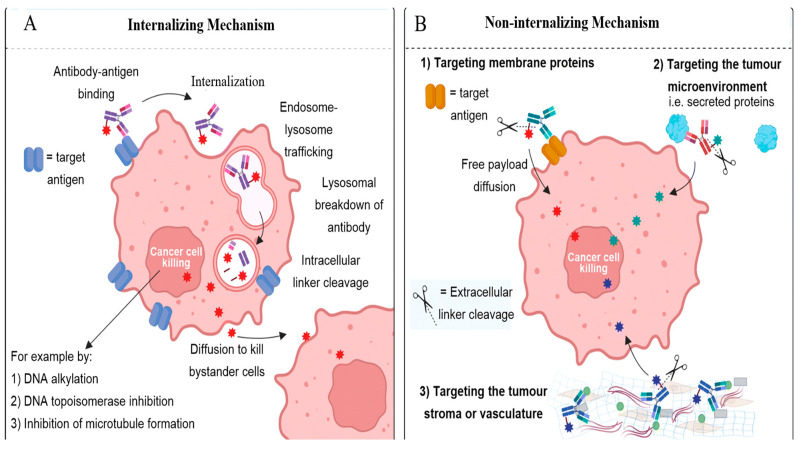
Internalizing versus non-internalizing mechanism. Adapted from in modified form [8].

**Figure 5 pharmaceutics-15-01242-f005:**
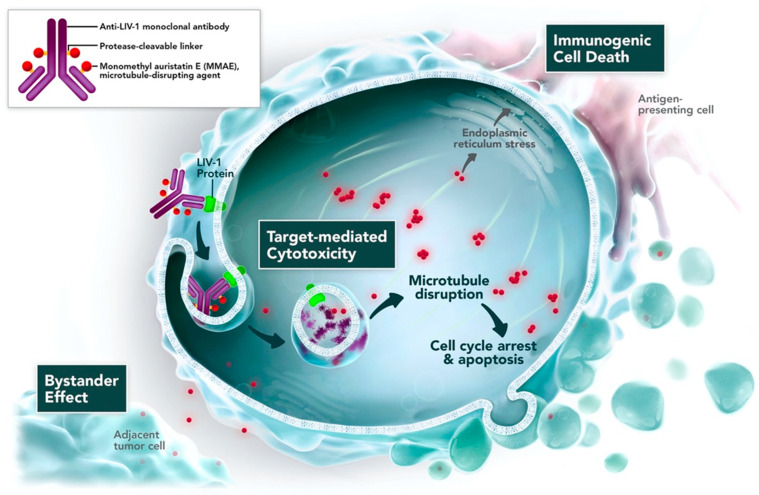
Mechanism of action of Ladrituzumab Vedotin. adapted from [92].

**Table 1 pharmaceutics-15-01242-t001:** ADC target antigens/proteins/stromal and vasculature components for the treatment of solid tumors.

Type	Categories of ADC Target Antigens or Other Targets	ADC Target Antigens or Other Targets
1	Well-established ADC target antigens	HER2, EGFR, EGFRvIII, c-MET, EGFR2, EGFR3
2	ADC target antigens overexpressed on cancer cells	EpCAM, BCMA, TROP-2, LIV-1, AXL, HER3, CD166, CEACAM5, GPNMB, Mesothelin, CD70
3	Non-internalizing ADC target cell-surface antigens	CD20, CD21, CD72, TAG72, CEACAM5, and NKA27
4	ADC target antigens in tumor microenvironment	CD25/IL2R, B7-H3, ANTXR1
5	ADC target antigens in cancer stem cells (CSCs)	PTK7, ROR1, 5T4
6	Targeting secreted proteins	Gal3BP, LRG1, and MMP9.
7	Targeting abundant stromal and vasculature components containing collagen, fibrin, fibronectin, and tenascin-C	Collagen, fibrin, fibronectin, and tenascin-C

**Table 2 pharmaceutics-15-01242-t002:** The pros and cons of internalizing and non-internalizing ADCs.

Internalizing ADCs	Non-Internalizing ADCs
Restricted choice of antigen targets	Broader choice of antigen targets
Necessitates overexpression of internalizing antigens	Circumvents inefficient internalization and trafficking to lysosomes
Cancer cells may acquire resistance related to internalization	Improved bystander effect and tumor penetration
May have limited cancer target	May be promising for targeting a wider range of cancers
Target antigens expressed on cancer cells	May target proteins other than those expressed in cancer cells, stromal and other tumor factors

**Table 3 pharmaceutics-15-01242-t003:** ADC drugs for breast cancer therapy either approved or in various phases of clinical trial.

ADC Drug	Target	Payload	DAR	Bystander Effect	Status	Adverse Effects	Ref.
Ado-trastuzumab emtansine	HER2	Matansine (Microdubule disrupting agent)	3–4	No	Approved in 2013, for HER2-positive MBC pretreated with trastuzumab and taxane (Adjuvant)	AST/ALT raises, thrombocytopenia, neuropathy	[2,6]
Trastuzumab deruxtecan	HER2	Deruxtecan (topoisomerase I inhibitor)	8	Yes	Approved in 2019 for HER2-positive MBC pretreated with trastuzumab and taxane	-	[2,6,105]
Trastuzumab duocarmazine	HER2	Duocarmycin prodrug	2.8	Yes	Not approved, phase III	Fatigue, conjunctivitis, dry eyes	[6,104]
Disitamab vedotin	HER2	MMAE	4	No	Not approved for breast cancer yet, in phase III. Approved for gastric or gastroesophageal junction cancer by NMPA (China)	Neutropenia, AST/ALT raises	[2,6,104]
Sacituzumab govitecan	TROP-2	SN-38 (topoisomerase I inhibitor)	7.6	Yes	Received accelerated approval for metastatic TNBC in 2020, and full approval in 2021. In 2023, received FDA approval for HER2- negative breast cancer	Neutropenia, anemia, diarrhea	[2,6,16,41,88]
Ladrituzumab vedotin	LIV-1 (TNBC)	MMAE	4	No	Not approved, phase III	Neutropenia, anemia	[2,6,104]
ARX-788	HER2	MMAF	1.9	–	Not approved, phase I/II	–	[2,104]
A166	HER2	Duostatin-5	N/A	–	Not approved, phase I/II	–	[2,104]
MRG002	HER2	MMAE	3.8	–	Not approved, phase I/II	–	[2,104]
ALT-P7	HER2	MMAE	2	–	Not approved, phase I	–	[2,104]
GQ1001	HER2	DM1	N/A	–	Not approved, phase I	–	[2,104]
SBT6050	HER2	Toll-like receptor 8 agonist (TLR8)	N/A	–	Not approved, phase I/II	–	[2]
ZW49	HER2 MBC	N-acyl sulfonamide auristatin	2	No	Phase I	-	[2]
MEDI4276	HER2	Tubulysin-based microtubule inhibitor	4	Yes	Phase I/II	-	[104]
XMT-1522	HER2	dolaflexin	12	-	Phase I/II	_	[6]
Patritumab deruxtecan	HER3	Deruxtecan (topoisomerase I inhibitor)	8	–	Not approved, phase I/II	–	[2]

## Data Availability

No supplementary data available.

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
