# Peer review of "Advances in Targeted Therapy of Breast Cancer with Antibody-Drug Conjugate"

_pharmaceutics, 2023, doi:10.3390/pharmaceutics15041242_

Round 1
Reviewer 1 Report
The authors introduced the currently approved ADC therapy and clinical trials for breast cancer. The information was well summarized, and the manuscript will be of interest to readers in the journal. However, the authors should address following comments and carefully edit the manuscript.
<Major points>
1. The abstract mainly provides general information of ADC. Since the title focuses on the breast cancer therapy, the author should include the information of breast cancer therapy including current success, problems, and future directions.
2. In current ADCs for breast cancer treatment, HER2 and TROP2 are important molecules. Therefore, the molecular functions and clinical significance should be explained in the text.
3. As the author described that ADC drug fam-trastuzumab deruxtecan-nxki (Enhertu) received FDA approval for unresectable or metastatic HER2-low breast cancer. This is thought to be a big progress in HER2 positive breast cancer treatment. Therefore, the author should explain the defference of the definition of HER2-positive (for trastuzumab monotherapy) and HER2-low (for trastuzumab-deruxtecan). How many patients will be covered or increased by the approval of trastuzumab-deruxtecan?
4. In Figure 3, the authors described several antibody formats such as IgG, Fab2 etc. In this review, the authors introduced many ADCs. Among them, are there any ADCs which contain the formats other than IgG? Or are there any promising ADC formats in future therapy?
5. In Figure 4A, is the internalization of ADC induced together with the target antigen? In the cartoon, the antigen was missing.
6. The internalization of ADC is a critical step to exert antitumor effects. However, it is not clear that the selection of target antigen or the selection of antibody are important for the efficient internalization of targets. Is there any information to answer the question?
7. The DAR (drug-antibody ratio) is different in each ADC. How is the DAR is controlled and optimized for the clinical use?
8. TNBC exhibits a poor prognosis, and novel modalities are desired. The authors described the approval of sacituzumab govitecan and clinical trial of Ladrituzumab Vedotin. The author listed the target antigens in Table 1. Are there any antigens targeting TNBC? At least, clinical trials of anti-GPNMB ADC were conducted previously (but failed). Are there other promising targets for TNBC?
9. Although the authors described many ADCs and the clinical trials, the current problems are not described clearly.
The authors mentioned “Systemic toxicity is one of the major factors for the failure of ADCs in the clinical setup”
What is systemic toxicity? How can the problem be resolved?
Are there more factors to be resolved?
Therefore, the author should clarify them and provide the possible solutions.
<Minor points>
10. The author should define abbreviations when they first come out.
<Abstract> TME
<Introduction> HER2 (page 2, line 3), TNBC (page 2, line 4), mAb (page 2, line 9)
<Figure 1 legend> FDA
<Figure 3> PEG
<Table 1> CSCs
<2. Internalizing and non-internalizing ADCs > PDAC (page 9, line 18)
<3.1.2 Trastuzumab deruxtecan> DAR (page 14, line 3), PDX (page 14, line 10), ILD (page 15, line 7)
<3.1.5 Sachituzumab Govitecan in HER-targeted breast cancer therapy> NCCN (page 18, line 9), TPC (page 18, line 12)
11. There are some misspellings in figures and tables.
Figure 4:
Internalising --- Internalizing
Internalisation--- Internalization
Table 2:
Monomethyla Aurastatin E --- Monomethyl Aurastatin E
Trop-2, TROP-2, and TROP2 are used in the text and table.
12. In Table 2, the target of Sacituzumab is TROP2, but not HER2 (Page 26).
Both Monomethyl Aurastatin E and MMAE were used in the table.
Trastuzumab deruxtecan and Patritumab deruxtecan are conjugated with same payload, Dxd. The author should unify the description of payload.
Furthermore, important references should be cited in the table.
13. The target of Ladrituzumab, LIV-1 is not listed in Table 1. Why?
14. In conclusion and future perspective,
“ADCs, either targeting human epidermal growth factor receptor or TNBC are prospective treatment approaches for breast cancer patients.”
--- Epidermal growth factor receptor is a target antigen, but TNBC is not target antigen.
Author Response
Reviewer 1
The authors introduced the currently approved ADC therapy and clinical trials for breast cancer. The information was well summarized, and the manuscript will be of interest to readers in the journal. However, the authors should address following comments and carefully edit the manuscript.
<Major points>
Thank you for your comments.
- The abstract mainly provides general information of ADC. Since the title focuses on the breast cancer therapy, the author should include the information of breast cancer therapy including current success, problems, and future directions.
Ans. Added some points in revision
- In current ADCs for breast cancer treatment, HER2 and TROP2 are important molecules. Therefore, the molecular functions and clinical significance should be explained in the text.
Ans. Explained in the text
- As the author described that ADC drug fam-trastuzumab deruxtecan-nxki (Enhertu) received FDA approval for unresectable or metastatic HER2-low breast cancer. This is thought to be a big progress in HER2 positive breast cancer treatment. Therefore, the author should explain the defference of the definition of HER2-positive (for trastuzumab monotherapy) and HER2-low (for trastuzumab-deruxtecan). How many patients will be covered or increased by the approval of trastuzumab-deruxtecan?
Ans. Definations are added.
The confirmed ORR based on ICR (Independent central review) in the 184 patients was 60.3% (95% CI: 52.9, 67.4) and the median DoR (Duration of response) was 14.8 months
- In Figure 3, the authors described several antibody formats such as IgG, Fab2etc. In this review, the authors introduced many ADCs. Among them, are there any ADCs which contain the formats other than IgG? Or are there any promising ADC formats in future therapy?
Ans. Addressed
- In Figure 4A, is the internalization of ADC induced together with the target antigen? In the cartoon, the antigen was missing.
Ans. Yes, blue color target antigen is shown in the figure.
- The internalization of ADC is a critical step to exert antitumor effects. However, it is not clear that the selection of target antigen or the selection of antibody are important for the efficient internalization of targets. Is there any information to answer the question?
Ans. Addressed in the text.
- The DAR (drug-antibody ratio) is different in each ADC. How is the DAR is controlled and optimized for the clinical use?
Ans. Described in the text
- TNBC exhibits a poor prognosis, and novel modalities are desired. The authors described the approval of sacituzumab govitecan and clinical trial of Ladrituzumab Vedotin. The author listed the target antigens in Table 1. Are there any antigens targeting TNBC? At least, clinical trials of anti-GPNMB ADC were conducted previously (but failed). Are there other promising targets for TNBC?
Ans. Trop-2, and LIV-1 are utilized for targeting TNBC. Yes, some of the target antigens listed in table 1 are suitable for targeting such as c-MET, EGFR, EpCAM, ROR1. We have addressed tis issues in our published TNBC papers. Currently, also working in this issue. However, their application in ADC drugs required further studies.
- Although the authors described many ADCs and the clinical trials, the current problems are not described clearly.
Ans. In some cases we have addressed this issue in revision.
The authors mentioned “Systemic toxicity is one of the major factors for the failure of ADCs in the clinical setup”
What is systemic toxicity? How can the problem be resolved?
Are there more factors to be resolved?
Therefore, the author should clarify them and provide the possible solutions.
Ans. Discussed in the text in revised version.
<Minor points>
- The author should define abbreviations when they first come out.
<Abstract> TME
<Introduction> HER2 (page 2, line 3), TNBC (page 2, line 4), mAb (page 2, line 9)
<Figure 1 legend> FDA
<Figure 3> PEG
<Table 1> CSCs
<2. Internalizing and non-internalizing ADCs > PDAC (page 9, line 18)
<3.1.2 Trastuzumab deruxtecan> DAR (page 14, line 3), PDX (page 14, line 10), ILD (page 15, line 7)
<3.1.5 Sachituzumab Govitecan in HER-targeted breast cancer therapy> NCCN (page 18, line 9), TPC (page 18, line 12)
Ans. Corrected.
- There are some misspellings in figures and tables.
Figure 4:
Internalising ---à Internalizing
Internalisation---à Internalization
Table 2:
Monomethyla Aurastatin E ---à Monomethyl Aurastatin E
Trop-2, TROP-2, and TROP2 are used in the text and table.
Ans. Addressed.
- In Table 2, the target of Sacituzumab is TROP2, but not HER2 (Page 26).
Both Monomethyl Aurastatin E and MMAE were used in the table.
Trastuzumab deruxtecan and Patritumab deruxtecan are conjugated with same payload, Dxd. The author should unify the description of payload.
Furthermore, important references should be cited in the table.
Ans. Addressed
- The target of Ladrituzumab, LIV-1 is not listed in Table 1. Why?
Ans. Has been listed.
- In conclusion and future perspective,
“ADCs, either targeting human epidermal growth factor receptor or TNBC are prospective treatment approaches for breast cancer patients.”
---à Epidermal growth factor receptor is a target antigen, but TNBC is not target antigen.
Ans. Modified

Reviewer 2 Report
The manuscript written by Subhan and Torchilin presents a very interesting role of targeted therapy of breast cancer with antibody drug conjugate. The idea and novelty of the manuscript is very unique. Therefore, this manuscript is of prime importance to the journal. The manuscript presents its idea in a very impeccable form. However, there are very minor flaw that need to be addressed before the manuscript can be accepted.
The manuscript should have been prepared in the journal`s template. Nevertheless, since the editorial board has accepted it, I assume they are fine with it.
Minor corrections needed:
In text where figure 1 is cited, there is reference 3. However, in the figure, reference 2 is cited. Please change accordingly if needed.
Please check the numbering of Table 1, Serial number 3 is repeated two times. Please change accordingly.
3.1 subsection is repeated twice. Change accordingly.
In the paragraph where trastuzumab deruxtecan (FDA approval in 2020) has been discussed, please change the dosage from 5.4 mg/km to 5.4 mg/kg.
Please also discuss the results of DESTINY-Breast02 trial as well.
Please add cross referencing for table 2.
The manuscript can be accepted after the said changes have been made.
Author Response
Reviewer 2
The manuscript written by Subhan and Torchilin presents a very interesting role of targeted therapy of breast cancer with antibody drug conjugate. The idea and novelty of the manuscript is very unique. Therefore, this manuscript is of prime importance to the journal. The manuscript presents its idea in a very impeccable form. However, there are very minor flaw that need to be addressed before the manuscript can be accepted.
Ans. Thank you.
The manuscript should have been prepared in the journal`s template. Nevertheless, since the editorial board has accepted it, I assume they are fine with it.
Ans. Currently, we are submitting as it is. Hopefully, journal will make it in Pharmaceutics format.
Minor corrections needed:
In text where figure 1 is cited, there is reference 3. However, in the figure, reference 2 is cited. Please change accordingly if needed.
Ans. Addressed
Please check the numbering of Table 1, Serial number 3 is repeated two times. Please change accordingly.
Ans. Corrected
3.1 subsection is repeated twice. Change accordingly.
Ans. Corrected.
In the paragraph where trastuzumab deruxtecan (FDA approval in 2020) has been discussed, please change the dosage from 5.4 mg/km to 5.4 mg/kg.
Ans. Changed
Please also discuss the results of DESTINY-Breast02 trial as well.
Ans, Discussed.
Please add cross referencing for table 2.
Ans. Added.
The manuscript can be accepted after the said changes have been made.

Reviewer 3 Report
Overall excellent review article
Page-10 Belantamab mafodotin does not have a cleavable linker. This is a non-cleavable type ADC consisting of MMAF. Please revise the language.
Page-12 Please summarise the pros/cons to compare non-internalizing and internalizing ADCs. A summary table will help.
Page-13 Also mention the current disadvantage of T-Dxd. For example, microphage-related toxicity remains a challenge.
Should bullet point separately approved ADC and ADCs in clinical trials
Author Response
Reviewer 3
Thank you for your comments.
Page-10 Belantamab mafodotin does not have a cleavable linker. This is a non-cleavable type ADC consisting of MMAF. Please revise the language.
Ans. Corrected
Page-12 Please summarise the pros/cons to compare non-internalizing and internalizing ADCs. A summary table will help.
Ans. Table added
Page-13 Also mention the current disadvantage of T-Dxd. For example, microphage-related toxicity remains a challenge.
Ans. Disadvantages of Approved ADCs added in the conclusion.
Should bullet point separately approved ADC and ADCs in clinical trials
Ans. Approved ADCs mentioned clearly (bold).
Round 2
Reviewer 1 Report
The manuscript is appropriately revised. However, careful checking of abbreviations and sentences is still required.
Introduction:
human epidermal growth factor receptor 2 positive, human epidermal growth factor receptor 2 positive (HER2+)
>>> "human epidermal growth factor receptor 2 positive" is redundant.
Abbreviations should be used uniformly.
ex) HER2, HER-2;
Trop-2, TROP-2 (TROP-2 is better because it is a human antigen in this review)
ISH should be spelled out, like IHC.
Author Response
Reviewer 1
Dear Reviewer
Thank you for minor corrections.
Introduction:
human epidermal growth factor receptor 2 positive, human epidermal growth factor receptor 2 positive (HER2+)
>>> "human epidermal growth factor receptor 2 positive" is redundant.
Ans. Corrected.
Abbreviations should be used uniformly.
- ex) HER2, HER-2;
Trop-2, TROP-2 (TROP-2 is better because it is a human antigen in this review)
Ans. Modified.
ISH should be spelled out, like IHC.
Ans. We have defined this in the text, ISH (in situ hybridization) such as Fluorescence in situ hybridization test (FISH). In case of HER2-negative it must be negative (ISH-).
Sincerely yours
Md Abdus Subhan
